# Exploring the Epidemiology of Melanocytic Tumors in Canine and Feline Populations: A Comprehensive Analysis of Diagnostic Records from a Single Pathology Institution in Italy

**DOI:** 10.3390/vetsci11090435

**Published:** 2024-09-14

**Authors:** Adriana Lo Giudice, Ilaria Porcellato, Giuseppe Giglia, Monica Sforna, Elvio Lepri, Maria Teresa Mandara, Leonardo Leonardi, Luca Mechelli, Chiara Brachelente

**Affiliations:** Department of Veterinary Medicine, University of Perugia, 06126 Perugia, Italy; adriana.logiudice@live.it (A.L.G.); giglia.giuseppe.93@gmail.com (G.G.); monica.sforna@unipg.it (M.S.); elvio.lepri@unipg.it (E.L.); maria.mandara@unipg.it (M.T.M.); leonardo.leonardi@unipg.it (L.L.); luca.mechelli@unipg.it (L.M.); chiara.brachelente@unipg.it (C.B.)

**Keywords:** melanocytic tumors, melanoma, canine, feline, epidemiology

## Abstract

**Simple Summary:**

Melanocytic tumors (MTs) are more prevalent in dogs than in cats. In dogs, they are the most frequent malignancy of the oral cavity, whereas in cats, ocular melanomas predominate. This study provides epidemiologic data (2005–2024) on MTs for both dogs and cats. Totals of 21128 canine and 4808 feline tumors were analyzed by the Veterinary Pathology Service of the Department of Veterinary Medicine (University of Perugia). Among these, 845 canine MTs (329 melanocytomas; 512 melanomas) were diagnosed, 485 from the skin, 193 from the oral mucosa, and 104 from the mucocutaneous junction. Older dogs were more likely to develop melanomas than melanocytomas (*p* < 0.001). In contrast, among feline tumors, only 60 were melanocytic (6 melanocytomas, 53 melanomas). Of these, 29 were cutaneous, 18 were ocular, and 9 were oral. In dogs, melanomas were more common in mucocutaneous locations than in cutaneous locations (*p* < 0.05); moreover, they were more common in the oral cavity compared to all other sites (*p* < 0.001). In cats, ocular melanomas were more common than cutaneous ones (*p* < 0.05). This study provides the prevalence of MTs in dogs and cats, supporting distinct epidemiological patterns of MTs, and confirming the significance of species-specific differences in the tumor prevalence, localization, and age distribution.

**Abstract:**

MTs are prevalent in dogs, representing the most frequent oral malignancy, compared to cats, in which ocular melanomas predominate. This study investigates the canine and feline MT epidemiology (2005–2024) of cases submitted to the Veterinary Pathology Service (University of Perugia). Among the canine neoplasms, 845 (4%) were melanocytic: 329 (39%) melanocytomas; 512 (61%) melanomas. Of these, 485 (57%) were cutaneous (4% of canine cutaneous neoplasms), 193 (23%) were oral (50% of oral canine neoplasms), and 104 (12%) were mucocutaneous. The average age of affected dogs was 10 years. Older dogs were more likely to have melanomas compared to melanocytomas (*p* < 0.001). There were 60 (1%) feline MTs: 6 (10%) melanocytomas; 53 (88%) melanomas. Of these, 29 (48%) were cutaneous (1% of feline cutaneous tumors), 18 (30%) were ocular, and 9 (15%) were oral (22% of feline oral tumors). The average age of affected cats was 11 years. In dogs, mucocutaneous melanomas were more common compared to cutaneous ones (*p* < 0.05); oral melanomas were more common compared to all other sites (*p* < 0.001). In cats, ocular melanomas were more common compared to cutaneous ones (*p* < 0.05). Our study provides the MT prevalence in a selected canine and feline population, revealing MT epidemiological patterns, highlighting species-specific differences in the tumor prevalence, localization, and age distribution.

## 1. Introduction

Melanocytic tumors are described in the human species and in domestic and non-domestic animals [1]. These tumors originate from melanocytes, which are dendritic neuroectodermal cells distributed in many different tissues of the body and specialized in melanin production for protective purposes [2,3]. Across all species, particularly in pets, there exists a distinction between melanocytoma, the benign form, and melanoma, the malignant counterpart [1,2,4,5,6]. In general, these tumors are typically black, when pigmented, or appear as whitish lesions in the absence of melanin production. These tumors can exhibit an exophytic growth pattern, or they may manifest as sessile nodules regardless of their site of origin [2,5]. The size of melanocytic tumors can range from less than a centimeter, primarily observed in melanocytomas, to several centimeters in the malignant counterpart, which often appear multilobulated and ulcerated [2,5,7,8]. Histologically, these tumors display different growth patterns. The cells can assume round to polygonal to spindle shapes, and they may be arranged in sheets, lobules, or bundles. Typically, benign melanocytic tumors are non-encapsulated but well demarcated, lacking infiltrative growth. Anisocytosis and anisokaryosis are mild, with a low number of mitoses. Conversely, melanomas are more invasive, exhibiting a moderate to marked cellular atypia and a higher number of mitoses [4,6,9,10,11]. The lack of pigmentation and the marked cellular atypia increase the difficulty of the diagnosis, broadening the spectrum of differential diagnoses. In these challenging cases, immunohistochemistry needs to be performed using specific melanocytic markers, such as Melan-A, PNL2, and Sox10, to confirm the diagnosis [12,13,14]. Among different species, including humans, melanomas demonstrate biologically aggressive behavior. Indeed, this tumor’s tendency towards local invasion and nodal and systemic metastasis leads to a generally unfavorable prognosis [4,5,9,11,14,15,16,17]. Due to these biological similarities between domestic animals and humans, and to the growing attention in pet care, in recent decades, the interest in these tumors also in veterinary medicine has increased. This is particularly evident for dogs, and to a lesser extent for cats [1,18,19,20]. In dogs, it has been highlighted that the majority of melanocytic tumors do not exhibit the same genetic features seen in humans, particularly in the cutaneous and UV-induced forms, which, in the human species, are almost always associated with the BRAF^V600E^ mutation [1,18,20,21]. Indeed, the cutaneous melanocytic tumors in dogs are mostly benign in contrast to their human equivalents, which display aggressive behavior. Additionally, canine digital melanocytic tumors appear to differ from those in humans. Indeed, it has been reported that RAS mutations are more frequent in canine digital melanomas compared to those in humans [22]. However, comparative studies have revealed that certain genes are mutated in both species, especially in oral melanomas, where humans and dogs share TP53 [23], MDM2 [24], NRAS [25], and PTEN [26] mutations, reflective of non-UV-driven initiating molecular events [27]. Therefore, canine oral melanoma is considered a good preclinical model for studying the human equivalent [18,25]. In feline species, genetic studies are few, and the most significant similarities to humans, in terms of tumor biological behavior, are seen in ocular melanomas. This type of melanoma is highly aggressive in both humans and cats, with a poor prognosis due to the lack of effective treatments [28]. Studies have been conducted to evaluate similarities between feline and human ocular melanomas, showing an increased expression of proteins such as KIT, BRAF, GNAQ, and GNA11, which play a crucial role in tumor progression in humans [29,30,31,32]. This raises considerations about the potential of the cat as a valuable preclinical model for studying uveal melanoma in humans. Despite this, epidemiological studies in veterinary medicine are few and date back to the early stages of melanoma research in domestic species, with a particular emphasis on dogs [33,34,35,36,37,38,39]. These reports indicate that cutaneous melanocytic tumors can arise everywhere on the integument and are relatively common in dogs, accounting for approximately 3–4% of all tumors [2]. This finding has been confirmed by more recent epidemiological studies on canine cutaneous tumors. For example, in Poland [40] and Korea [38], melanomas and melanocytic tumors represented 4.65% and 4.23% of all cutaneous tumors, respectively. In Germany, melanocytic tumors represented 5.2% of all diagnosed neoplasms in dogs, and the most prevalent location was the integument (92%), where 35.4% of the tumors were malignant [33]. However, incidence rates vary geographically, with lower figures reported in the UK (13 cases per 100,000 dogs annually) [35], and slightly higher rates in regions like Tulsa and Alameda (9% and 7% of neoplastic lesions, respectively) [41]. Overall, melanocytic tumors are a significant health concern for dogs, ranking as the sixth most frequent neoplasm in a Swiss study [42]. Moreover, it is generally described that melanocytomas are more common in the skin [2], as confirmed in an epidemiologic study in Greece [36], but the aforementioned Swiss study revealed that, in their registry, almost two-thirds of melanocytic tumors were malignant [42]. This trend was also observed in Poland, where melanocytomas made up 1.37% compared to 4.65% of melanomas [40]. Within the subgroup of digital neoplasms, melanocytic tumors are the second most frequent type of neoplasia and are often aggressive [43]. Canine oral melanocytic tumors are found in the gingiva, lips, tongue, and hard palate [8]. Oral melanomas are considered the most frequent malignant tumor in dogs, accounting for about 30–40% of oral tumors [8,10,14]. Regarding ocular melanoma, uveal melanocytic neoplasms have been reported to represent the majority of primary intraocular neoplasms (70%), with intraocular melanocytomas being more common than melanomas [44]. While no specific gender predisposition has been demonstrated so far [2,14,39], certain breeds appear to be more prone to the development of melanocytic tumors, particularly those with darker coats and skin. Some studies report that Scottish, Airedale, and Boston Terriers, Miniature and Standard Schnauzers, Doberman Pinschers, Vizslas, Golden and Chesapeake Bay Retrievers, Irish Setters, Chow Chows, Boxers, and Cocker and Springer Spaniels are at higher risk [2,14,33]. More recent analyses have also identified breeds like Rottweilers, Rhodesian Ridgebacks, Labrador Retrievers, and Pekingese/Poodle mixed breeds as being highly represented [33,42,43]. Melanocytic tumors typically arise in older dogs, with an average age ranging from 8 to 11 years, although younger dogs can also be affected [2,39,45]. In cats, melanocytic tumors are less common, with ocular variants, particularly those affecting the iris, being the most frequently observed [46]. Cutaneous forms, often originating from the pinnae, and oral melanocytic tumors are notably rare, with melanocytomas being very uncommon [2]. Dark-haired (black or gray) cats are more susceptible [2]. The reason for the feline species predisposition to ocular melanomas remains largely unknown. Also, in feline species, no gender predisposition is reported, and the average age for the development of a melanocytic tumor ranges between 8 and 12 years [2,9]. Epidemiologic data on melanocytic tumors in Italian dogs and cats are limited. For this reason, a comprehensive epidemiological investigation is needed to understand their distribution and demographic correlations in the canine and feline populations. This study aims to provide epidemiological data on melanocytic tumors in dogs and cats from a single Italian institution.

## 2. Materials and Methods

### 2.1. Data Collection

Data were obtained from the electronic archives of the Service of Veterinary Pathology of the Department of Veterinary Medicine at the University of Perugia. The dataset utilized in this study comprised cytologic samples, surgical biopsy cases, and necropsy cases spanning from January 2005 to February 2024. All cases with a diagnosis of neoplasia were included in the study, while hyperplastic and preneoplastic lesions were excluded from the dataset. To export all the diagnoses and the data from each case, we used the keywords “neoplasia”, “tumor”, and “neoformation”. Subsequently, distinct subgroups were delineated for cutaneous and oral tumors, including those originating from the dental and periodontal tissue, tongue, and tonsils. Furthermore, melanocytic tumors in dogs and cats were sorted and extrapolated from the archive using the keywords “melanocytic tumor”, “melanoma”, and “melanocytoma”. The cases with no confirmed diagnosis (e.g., when the melanocytic tumor diagnosis was just included in the differentials) were excluded from the study. Data on melanocytic tumor samples, including sex, age, diagnostic code, species, and breed, were extracted from the database and are summarized in the Appendix A. Unfortunately, not all cases included in the study had complete anamnesis data available. The diagnosis of melanocytic tumors in our institution is made following the histologic and immunohistochemical guidelines, and the samples are evaluated by board-certified pathologists [5,13].

### 2.2. Statistical Analyses

Statistical analyses were conducted using Jasp software (version 0.18.3). Initially, descriptive statistics were examined to understand the basic characteristics of the dataset, focusing on metrics. Histograms were generated to assess the normality of the data distribution. The mean age and standard deviation were reported for dogs (normally distributed), while the median age and 95% confidence interval (95% CI) were used for cats (skewed distributed data). To compare age means between tumor types (melanocytoma vs. melanoma), Student *t*-tests (dogs) and Mann–Whitney tests (cats) were employed. The association between the tumor localization and tumor type was assessed using the Chi-square test, with the odds ratios calculated to quantify the likelihood of tumor development in specific sites. Finally, Fisher’s exact test was used to examine any association between the tumor types and breeds.

## 3. Results

Data from our archives included 21128 canine and 4808 feline tumors submitted to our laboratory from January 2005 to February 2024.

### 3.1. Canine Melanocytic Tumors

Among the canine tumors, 10930 (52%) were cutaneous and 381 (2%) were from the oral cavity. Out of the total canine tumors, 845 were identified as melanocytic tumors (4%), submitted to our laboratories mostly from Umbria and Lazio (Figure 1). A total of 46 cases needed immunohistochemistry using anti Melan-A, PNL2, and Sox10 antibodies to confirm the diagnosis. Specifically, there were 329 (39%) melanocytomas, 505 (61%) primary melanomas, 7 metastatic melanomas (with no primary tumor sampled) (0.8%), and 4 (0.4%) melanocytic tumors diagnosed without further specific indication. In 485 dogs, melanocytic tumors had a cutaneous localization (57%; 4% of all canine cutaneous tumors); specifically, 33 were digital, while in 193 dogs, tumors occurred in the oral cavity (23%; 50% of all canine oral neoplasia). Additionally, among the tumors arising from the mucocutaneous junction, 60 cases (7%) were from the lip and 44 cases (5%) were from the eyelid (Table 1). Furthermore, 49 cases (6%) were ocular, and 4 cases (0.4%) were from other mucosal epithelia (three nasal and one vulvar). Metastases in different sites were observed in 10 cases (1%).

The analysis of the tumor localization and the comparison between the melanocytoma and melanoma groups showed significant differences (*p* < 0.001). In particular, there was a significant association between the tumor type and mucocutaneous and cutaneous sites (*p* < 0.05), indicating that melanomas were more common in mucocutaneous locations compared to cutaneous locations, while melanocytomas were more common on skin rather than on mucocutaneous sites; other associations were found between cutaneous and oral sites (*p* < 0.001), between mucocutaneous and oral sites (*p* < 0.001), and between oral and ocular sites (*p* < 0.001), indicating that melanomas were more common in the oral cavity compared to the mucocutaneous junctions and eyes. Regarding the demographics of the dogs with melanocytic neoplasia, 329 were female (39%) and 471 (56%) were male. In 45 cases (5%), the sex was not reported by the referring vet. The neutering status was not included because only a small portion of the examined population had this information specified in their medical history. Regarding the type of tumor, 196 (23%) males had a diagnosis of melanocytoma, while 272 (32%) males had a diagnosis of melanoma; 120 (14%) females had a diagnosis of melanocytoma, while 209 (24%) females had a diagnosis of melanoma. No significant association was found between the sex and diagnosis (melanoma vs. melanocytoma) (*p* > 0.05). The mean age was 10 years (sd ± 3.16). The youngest dogs diagnosed with a melanocytic tumor were two 1-year-old male and female German Shepherds with a uveal melanocytoma and palpebral melanoma, respectively, while the oldest was a 21-year-old female English Setter, also diagnosed with a uveal melanocytoma. The mean age of dogs with melanocytomas was 9 years (sd ± 3), whereas the mean age of dogs with melanomas was 11 years (sd ± 3) (Figure 2).

The comparison between the age means of the melanocytoma and melanoma groups showed a significant difference (*p* < 0.001), with a higher mean age in dogs with melanomas compared to dogs with melanocytomas. Age data were not available for 93 patients (11%). Of the total dogs diagnosed with melanocytic tumors, 32% were mixed-breed (268 cases). Among the pure breeds, the most common were German Shepherds (41 cases, 4.7%); Rottweilers (36 cases, 4.8%); Labrador Retrievers (35 cases, 4.1%); Golden Retrievers (33 cases, 3.9%); Dachshunds (31 cases, 3.7%); Cocker Spaniels (28 cases, 3.1%); Miniature Pinschers (28 cases, 2.9%); Boxers (24 cases, 2.9%); Yorkshire Terriers (22 cases, 2.6%); Giant Schnauzers (18 cases, 2.1%); Dobermann Pinschers (15 cases, 1.7%); English Setters (15 cases, 1.7%) (Table 2). Breed information was not reported for 66 dogs (7.8%). The comparison between the breeds and the melanocytoma and melanoma groups showed no significant differences (*p* > 0.05).

### 3.2. Feline Melanocytic Tumors

In contrast, considering all the feline tumors, 1828 (38%) were cutaneous and 68 (1%) were from the oral cavity. Out of these, 60 were identified as melanocytic tumors, representing 1% of the total feline tumors, including 6 (10%) melanocytomas and 53 (88%) melanomas, with one case diagnosed as a melanocytic tumor, without further specific indication. A total of 14 cases needed immunohistochemistry against Melan-A, PNL2, and Sox10 to confirm the diagnosis. Similar to the dogs, most of the samples were submitted from Umbria and Lazio (Figure 3).

A total of 29 cases of feline melanocytic tumors were cutaneous (48%; 1% of cutaneous neoplasia), followed by 18 cases of ocular tumors (30%, most of which arose from the iris), 9 cases of tumors arising from the oral cavity (15%; 22% of the oral tumors), and 1 case each from the nasal mucosa and lip (Table 3). Additionally, there were two metastatic melanocytic tumors (3%), one from the spleen and one from the liver, presented without evidence of a primary lesion.

The comparison between the tumor localization and the melanocytoma and melanoma groups showed a significant difference. In particular, melanomas were the most represented in the ocular location compared to the cutaneous one (*p* < 0.05). Cats with melanocytic tumors were 27 (45%) females and 28 (47%) males; for five patients (8%), the sex was not disclosed by the referring vet. The neutering status was not included because only a small portion of the examined population had this information specified in their medical history. Regarding the type of tumor, 3 (5%) males had a diagnosis of melanocytoma, while 24 (32%) males had a diagnosis of melanoma; 2 (3%) females had a diagnosis of melanocytoma, while 24 (32%) females had a diagnosis of melanoma. No significant association was found between the sex and diagnosis (melanoma vs. melanocytoma) (*p* > 0.05). As for the breed, 36 (60%) of the cats were domestic shorthairs, 3 (5%) were Persian cats, and for 21 (35%) of the patients, the breed information was unavailable (Table 4). The analysis conducted on the possible relation between the breed and a diagnosis of melanocytoma and melanoma did not highlight any significant differences (*p* > 0.05).

The median age was 12 years (95% CI: 10–12.3). The youngest cats diagnosed with a melanocytic tumor were 2 years old, a male and a female, domestic shorthairs with diffuse iris melanomas, while the oldest was a 19-year-old male domestic shorthair with a cutaneous melanoma. Age information was not available for 15 patients (32%). The median age of cats with melanocytomas was 11 years (95% CI: 8–13.9), whereas the median age of cats with melanomas was 12 years (95% CI: 9.9–12.4) (Figure 4).

The comparison between the age means of the melanocytoma and melanoma groups did not highlight any significant differences (*p* > 0.05).

## 4. Discussion

In this study, conducted on samples received in our laboratory from various Italian regions over a 19-year period, we evaluated the frequency and characteristics of melanocytic tumors in dogs and cats. Our findings reveal a higher prevalence of melanomas compared to melanocytomas in both species, in agreement with previously reported data [40]. This was particularly evident in cats, where only 6 melanocytomas were identified, and in canine oral melanomas, where only 10 melanocytomas were diagnosed. Additionally, in our dataset, the number of canine cutaneous melanomas (228) is slightly lower compared to that of melanocytomas (253). This contrasts with some reports in the scientific literature, which indicate a ratio of melanocytomas to melanomas of 2:1, or even 4:1 in certain studies [2]. In Germany, it has been reported that melanomas on the skin account for 35.4% of all cutaneous melanocytic tumors [33]. However, as previously mentioned, more recent epidemiological analyses have reported different findings. For instance, a study conducted in Switzerland showed that cutaneous melanomas represented about three-quarters of all enrolled melanocytic tumors [42]. Similarly, another study from Poland indicated that cutaneous melanomas accounted for 4.65%, while melanocytomas represented only 1.37%, of all melanocytic tumors in their dataset [40]. In general, it is important to highlight that single-institution studies can be influenced by geographic trends, which may introduce some bias. Despite this, our extensive dataset provides detailed and consistent diagnostic records, thereby offering valuable insights into the epidemiology of these tumors in Italy. Although many of our findings align with those of other studies, as previously mentioned, the unusually high number of cutaneous melanomas in our sample suggests the need for collaboration with other Italian institutions. Expanding this study to include additional cases from different regions would allow for the collection of more comprehensive epidemiological data, helping to refine our understanding of the prevalence and characteristics of these tumors in the broader canine and feline populations. As in other studies, we did not identify a significant association between the tumor type and breed or sex in either species [5].

Furthermore, in both dogs and cats, our data confirm that melanocytic tumors are more common in older animals, with relatively few cases occurring in young patients. This age distribution is in agreement with what is reported in other studies [5,8,47,48,49]. Specifically, dogs with melanomas tend to be older than those with melanocytomas. The onset of the malignant form of this tumor in older age might suggest that the tumorigenesis and tumor progression are linked to underlying age-related mechanisms. This correlation may indicate that, as animals age, cumulative genetic mutations, diminished immune surveillance, and other age-associated biological processes contribute to the development and aggressiveness of malignant tumors. One more consideration could be that the delay between the tumor onset and its diagnosis can be significant, as owners may not detect tumors early, especially in dogs with long and thick coats or in less visible areas, such as the oral cavity or eyes. This diagnostic delay may have an impact on the correct assessment of the actual age at which these tumors develop, particularly for the slow-growing ones. Moreover, in our study, we did not statistically compare the ages across different breeds because certain breeds are more long-lived than others, and this aspect could have highlighted significant differences. However, our dataset is predominantly composed of mixed-breed dogs, with relatively few purebreds represented. This imbalance could compromise the validity of any analysis based on breed-specific longevity. For this reason, further studies involving more institutions are necessary to investigate this aspect in a more robust manner.

In our cohort, certain breeds, including Rottweilers, Labrador Retrievers, Golden Retrievers, Dobermann Pinschers, and Schnauzers, were overrepresented with melanocytic tumors, consistent with previous reports [50]. Although our study did not reveal a significant association between the breed and tumor type, this finding should be interpreted with caution due to the relatively small sample size within specific breed subgroups, as evidenced by the high proportion of mixed-breed dogs in our cohort. Larger studies with more balanced breed representation are needed to confirm these findings.

Additionally, we found a significant association between the tumor localization and the type of melanocytic tumor. In dogs, there is an association between mucocutaneous and cutaneous sites and the tumor type, indicating that melanomas are less likely to be found in the mucocutaneous location compared to melanocytomas. Other associations between cutaneous and oral sites, mucocutaneous and oral sites, and oral and ocular sites indicate that melanomas are more common in the oral cavity compared to the mucocutaneous and ocular locations. As noted by Esplin et al., this could be because a subset of tumors arising from the mucous membranes of the lips and oral cavity tend to be well differentiated, exhibiting lower rates of progression compared to their more aggressive counterparts. This finding suggests that some melanocytic tumors in these locations may follow a more indolent course [51]. In contrast, in cats, there is a significative association between the type of tumor and cutaneous and ocular sites, showing that ocular melanomas are the most represented compared to the cutaneous ones. This result is consistent with the scientific literature [28]; however, the reasons why this species is predisposed to ocular melanomas remain largely unknown. While studies have observed this predisposition, the underlying mechanisms, whether genetic, environmental, or linked to anatomical or physiological factors, have yet to be fully elucidated. These associations suggest that the benign or malignant nature of the tumor may depend on its localization, both in dogs and cats. As we also know from the literature, these results may have important implications for the diagnosis, prognosis, and treatment of canine and feline melanocytic tumors.

Regarding the diagnostic approaches, as mentioned before, regular oral examinations in dogs and thorough ocular inspections in both dogs and cats is difficult for owners but could facilitate the earlier detection of melanomas, which could improve the therapeutic approach and, finally, the prognosis.

Further studies should aim to validate these findings across multiple institutions and geographic regions to investigate the impact of potential regional variation, for example, pollutants and environmental differences, and, at the same time, enhance the generalizability of the results.

Furthermore, the relationship between the tumor type, age, and localization suggests that further investigations should be aimed at the understanding of the underlying biological pathomechanisms. Research into genetic, environmental, and lifestyle factors that influence the development and progression of melanocytic tumors in pets could lead to more effective prevention and management strategies.

## 5. Conclusions

This study provides further data on the epidemiology of melanocytic tumors in dogs and cats, highlighting the predominance of melanomas, the association with age, and the significance of developing a benign or malignant melanocytic tumor based on its localization. These findings could have important implications for veterinary practice and serve as a preliminary step for future research aimed at improving the outcomes of dogs and cats affected by these tumors.

## Figures and Tables

**Figure 1 vetsci-11-00435-f001:**
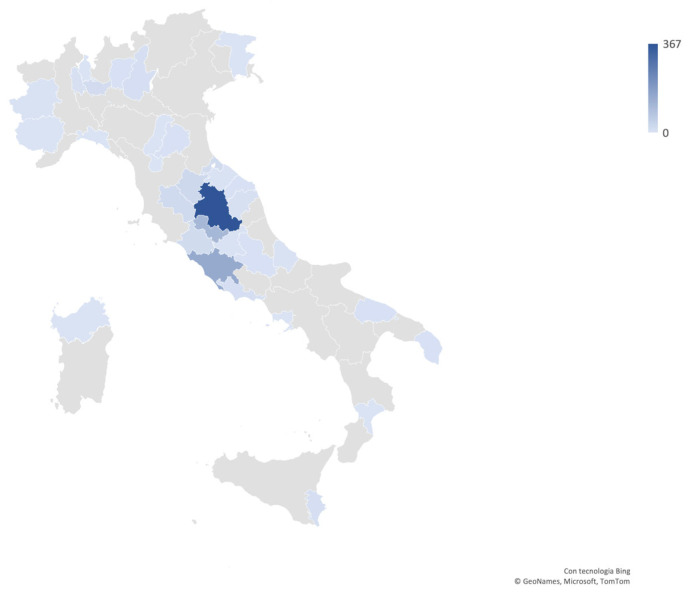
Density map representing the areas from where canine melanocytic tumor samples were submitted to our laboratory. The majority of samples were submitted by practitioners from Umbria and Lazio, and a minority of samples came from other Italian regions.

**Figure 2 vetsci-11-00435-f002:**
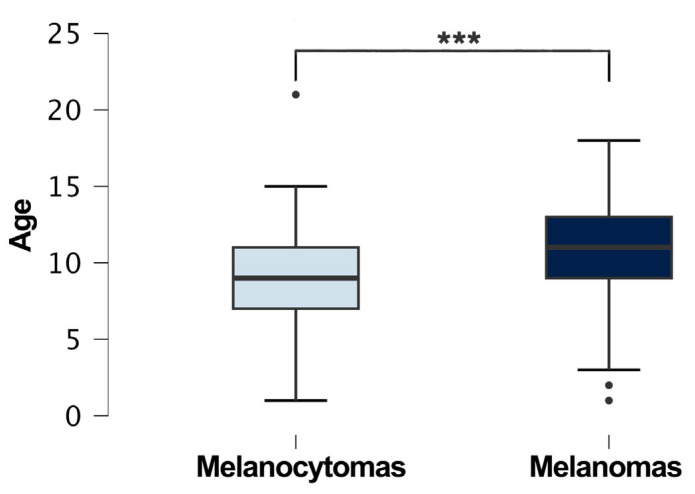
The box plot illustrates the age distribution of dogs diagnosed with melanocytomas and melanomas. Dogs with melanocytomas (287 cases) have a mean age of 9 years, while those with melanomas (462 cases) have a mean age of 11 years. A few outliers are present, with melanocytomas having one older outlier and melanomas showing several younger outliers. The comparison between these two tumor types and the age has a significant difference (*** *p* < 0.001).

**Figure 3 vetsci-11-00435-f003:**
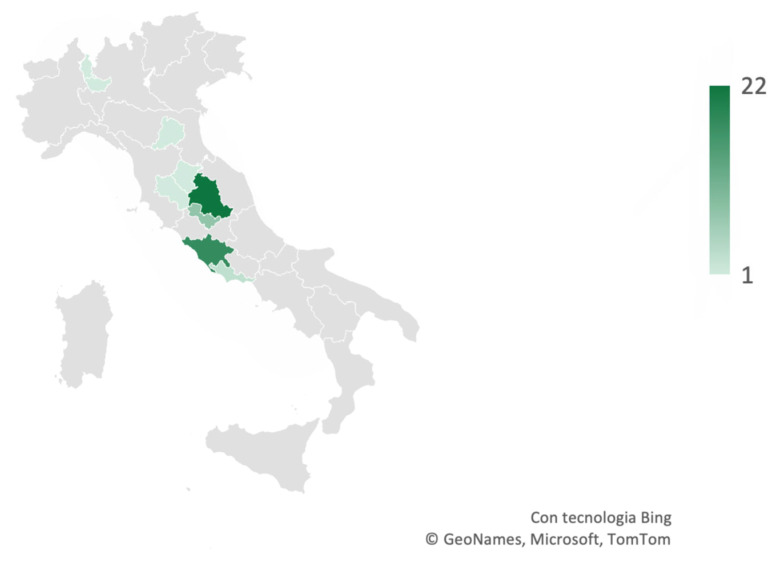
Density map representing the areas from where feline melanocytic tumor samples were submitted to our laboratory. The majority of samples were submitted by practitioners from Umbria and Lazio, and a minority of samples came from other Italian regions.

**Figure 4 vetsci-11-00435-f004:**
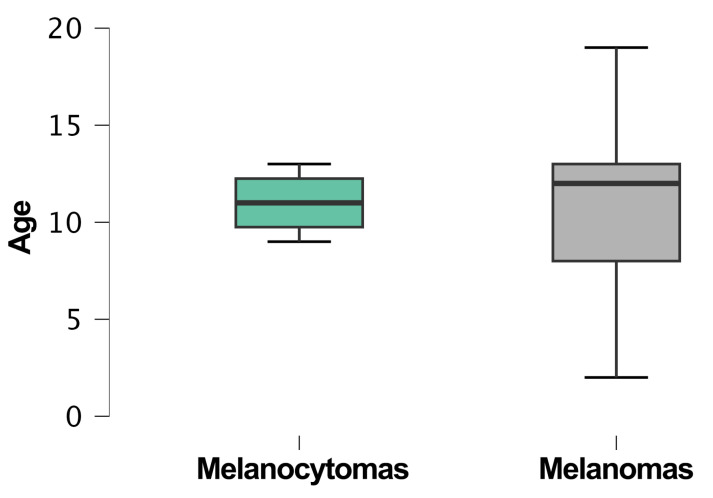
The box plot illustrates the age distribution of cats diagnosed with melanocytomas and melanomas. Cats with melanocytomas (6 cases) had a median age of 11 years, while those with melanomas (49 cases) had a median age of 12 years.

**Table 1 vetsci-11-00435-t001:** The table shows the numbers of cases of canine melanocytomas and melanomas and their percentages in the most represented tumor locations.

Location	Melanocytomas	Melanomas	Percentage *
Cutaneous	253	228	57%
Mucocutaneous	43	61	12%
Oral	10	182	23%
Ocular	23	26	6%

* The percentage refers to the total number of melanocytic tumors.

**Table 2 vetsci-11-00435-t002:** The table shows the numbers of cases of canine melanocytomas and melanomas and the principal localizations in the most represented breeds in our study.

Breed	Melanocytomas					Melanomas				
	Cutaneous	Mucocutaneus	Oral	Ocular	Total	Cutaneous	Mucocutaneus	Oral	Ocular	Total
Mixed Breed	81	17	1	6	105	66	23	61	7	157
German Shepherds	11	2	1	5	19	12	4	2	3	21
Rottweilers	7	-	1	-	8	16	2	8	-	26
Labrador Retrievers	8	-	1	1	10	18	2	5	-	25
Golden Retrievers	7	1	-	3	11	6	4	9	1	20
Dachshunds	10	2	2	1	15	9	1	6	-	16
Cocker Spaniels	10	2	-	-	12	2	6	7	-	15
Miniature Pinschers	8	-	-	-	8	15	-	2	-	17
Boxers	8	2	-	1	11	9	-	2	2	13
Yorkshire Terriers	10	-	-	-	10	6	1	4	-	11
Dobermann Pinschers	5	1	-	-	6	7	1	1	-	9
Giant Schnauzers	7	-	-	-	7	9	1	1	-	11
English Setters	1	1	2	1	5	2	-	7	1	10
Other Breeds *	57	14	2	4	77	37	14	47	10	108
Breeds n/a **	23	1		1	25	14	2	20	2	38
Total					329					497

* All breeds with fewer than 10 cases of melanocytic tumors were grouped together under a single category labeled “Other Breeds”. ** n/a stands for “not available”.

**Table 3 vetsci-11-00435-t003:** The table shows the numbers of cases of feline melanocytomas and melanomas and their percentages in the most represented tumor locations.

Location	Melanocytomas	Melanomas	Percentage *
Cutaneous	6	22	48%
Oral	0	9	30%
Ocular	0	18	15%

* The percentage refers to the total number of melanocytic tumors.

**Table 4 vetsci-11-00435-t004:** The table shows the numbers of cases of feline melanocytomas and melanomas in domestic shorthair and Persian cats.

Breed	Melanocytomas				Melanomas			
	Cutaneous	Oral	Ocular	Total	Cutaneous	Oral	Ocular	Total
Domestic Shorthair	4	-	-	4	13	5	11	29
Persian	-	-	-	-	-	-	3	3
Breeds n/a **	2	-	-	2	9	4	4	17
Tot				6				49

** n/a stands for “not available”.

## Data Availability

The dataset containing signalment (Breed, sex, age) and diagnosis are available in the Appendix A.

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
