# Peer review of "Exploring the Epidemiology of Melanocytic Tumors in Canine and Feline Populations: A Comprehensive Analysis of Diagnostic Records from a Single Pathology Institution in Italy"

_vetsci, 2024, doi:10.3390/vetsci11090435_

Round 1

Reviewer 1 Report

Comments and Suggestions for Authors

The authors have compiled descriptive statistics on melanocytic tumors in the material submitted to a pathological institution in Italy. Unfortunately, the evaluation is superficial, and the interpretation shows considerable gaps, even when other literature is included.

Here are my comments:

Title: it should read "single PATHOLOGY institution IN ITALY"

Line 29: delete "to"

Introduction: The literature is sometimes cited unprecise and superficially. It is unclear what the current state of the literature on the epidemiology of melanocytic tumors in dogs and cats is (e.g., from which countries and which institutions does the data originate?) and why the present study is necessary or valuable. What is the hypothesis of the study?

Line 58: Why do reduced pigmentation and strong cell atypia represent a difficulty in diagnosis? In the case of differential diagnoses, an immunohistochemical examination (e.g., Melan A) is available - this should be referred to here. You did not mention how often this was a problem in your study – how often was immunohistochemistry necessary? – if this is not part of the study – you can delete it from the introduction.

Line 64: The genetic similarities of melanomas in dogs and cats with those in humans are relatively low because the melanomas in dogs and cats do not have a BRAF mutation! It has also been shown that RAS mutations are much more common in dogs than in humans, at least in melanomas of the toe (Conrad et al.). Generalities are written here that do not show that the literature has been read and understood in detail.

Line 80: Why is an update of the literature important for such a small area in Italy? And can we speak of an update when the study spans almost 20 years? After all, no development of the tumor numbers over time was established.

Material and methods

How did you generate the data, e.g., electronica archive? How did you homogenize/standardize the data? What were the inclusion and exclusion criteria for the cases in your study?

Line 107: Regarding the evaluations of age: Either the age is normally distributed - in which case the mean value is given with standard deviation, or the age is not normally distributed, in which case the mean value is shown with median - minimum-maximum. The statistics are confusing. The evaluation of breeds is extremely minimalistic.

Results:

Figure 1+2: Ultimately, the figures only show where the samples come from geographically in Italy. No evaluation or conclusions were drawn from the distribution. The figures are, therefore, superfluous.

Table 1: The proportion of 228 melanoma in the skin is surprisingly high. Usually, melanocytic tumors of the skin are benign (melanocytoma). At no point has it been explained why the proportion of melanomas in the skin is so high in this study? What classification/grading is used in the pathological institution? à Mat&Met?

On the other hand, the proportion of melanocytomas in the mucocutaneous area is very high. Furthermore, it is not described whether these are mucocutaneous transitions in the mouth or, for example, in the genital tract. There is also no more precise information on the cutaneous localization (e.g., back, limb).

Lines 140-148: The evaluation of the genders is unsystematic and confused. Here, all melanocytic tumors are evaluated according to gender - but without any indication of significance. The distribution of gender in melanomas or melanocytomas is given without numerical information.

Line 149: 3.16 (point instead of comma)

Line 150: Is the tumor of the 1-year-old dog a melanoma or a melanocytoma? Where was it localized, and what kind of dog was it? The same questions arise for the tumor of the oldest dog. And later, the same for the cats...

Line 151: See Material & Methods for the age data

Figure 2: Are there statistically significant differences between the melanocytomas and melanomas? The representation in the boxplots refers to medians/quartiles; the text refers to mean and standard deviation - a wild mixture of different statistical data.

Line 156: The heading is in the wrong place

Lines 160-166: The evaluation of breeds is only descriptive, and no reference is made to the overall population; therefore, no breed disposition can be read from this. It would be helpful to indicate the orthopaedic Joos in relation to the total population.

Are there statistically significant differences between melanocytomas and melanomas? (races, age, gender)? This is not clearly elaborated.

Melanocytic tumors in cats: There are different stages and classifications of ocular melanocytic tumors in cats, which are not discussed in detail here. However, this would be desirable.

Information on age, youngest, and oldest cat as for dogs.

Melanocytic tumors of the cat: There are different stages and classifications specifically for ocular melanocytic tumors of the cat, which are not discussed in detail here. However, this would be desirable.

Information on age, youngest and oldest cat as for dogs

Do you only have DSH and Persians in Italy???? – Which breeds are in your total population of pathology samples?

Discussion:

Line: The authors write that this study would provide valuable insights into the epidemiology of the tumors. However, which information from the present survey has what value and why? - Especially as the authors note that almost all known publications come to the same results.

Line 225: It was previously stated that there is no significant age difference between dogs with melanoma and melanocytoma. With a standard deviation of 3 years and an age difference of 9 to 11 years, one cannot speak of a trend. Furthermore, the effects of the breed were not taken into account. To what extent do melanomas or melanocytomas occur more frequently or less frequently in certain breeds - some breeds have a longer life expectancy than others. The data is interpreted very uncritically here.

Furthermore, looking at the breeds, the hair coat color should be considered!

Line 232 ff: Only results are repeated here, and there is no critical examination of the literature or the author's data. There is no indication of limitations and no explanation of what the study really brings that is new.

Line 243: The regular oral cavity examination is not a conclusion from the present work! - This is generally known because previous studies and experiences are already available.

Line 245 ff: So far, no recognizable regional differences between the results of the own study from Italy compared to other studies have been identified, so it is questionable why further studies should be conducted with others in catchment areas and regions. The possible predisposing factors, such as UV light, genetic factors, racial predisposition, racial distribution in the different countries, etc., are not discussed.

Conclusions

Line 258: The conclusion that the results of this study have any impact on prognosis or veterinary practice cannot be drawn from the results presented in this study.

Author Response

The authors have compiled descriptive statistics on melanocytic tumors in the material submitted to a pathological institution in Italy. Unfortunately, the evaluation is superficial, and the interpretation shows considerable gaps, even when other literature is included.

We are truly grateful, for the evaluation given by the reviewer. Their valuable feedback and suggestions have improved the quality and clarity of our document. Following are the responses to reviewer’s comments:

Here are my comments:

Title: it should read "single PATHOLOGY institution IN ITALY"

The title has been changed

Line 29: delete "to"

The “to” has been deleted

Introduction: The literature is sometimes cited unprecise and superficially. It is unclear what the current state of the literature on the epidemiology of melanocytic tumors in dogs and cats is (e.g., from which countries and which institutions does the data originate?) and why the present study is necessary or valuable. What is the hypothesis of the study?

The section of the introduction addressing epidemiology has been carefully revised to ensure accuracy and clarity (lines 89-139). Additionally, the aim of the study has been clarified.

Line 58: Why do reduced pigmentation and strong cell atypia represent a difficulty in diagnosis? In the case of differential diagnoses, an immunohistochemical examination (e.g., Melan A) is available - this should be referred to here. You did not mention how often this was a problem in your study – how often was immunohistochemistry necessary? – if this is not part of the study – you can delete it from the introduction.

An explanation about immunohistochemistry has been added in the introduction. The result section has been implemented with immunohistochemical information (lines 60-62).

Line 64: The genetic similarities of melanomas in dogs and cats with those in humans are relatively low because the melanomas in dogs and cats do not have a BRAF mutation! It has also been shown that RAS mutations are much more common in dogs than in humans, at least in melanomas of the toe (Conrad et al.). Generalities are written here that do not show that the literature has been read and understood in detail.

The paragraph has been implemented and improved (lines 68-79).

Line 80: Why is an update of the literature important for such a small area in Italy? And can we speak of an update when the study spans almost 20 years? After all, no development of the tumor numbers over time was established.

Because of the lack of data reported in Italy about melanocytic tumor epidemiology, this study aims to be an observational analysis of the situation in a portion of our country. Additionally, our institution belongs to a network of Italian institutions that are collecting data on canine tumors; therefore, the long-term objective of this network is to describe the data with all the other institutions to evaluate the potential epidemiologic significance.

Material and methods

How did you generate the data, e.g., electronica archive? How did you homogenize/standardize the data? What were the inclusion and exclusion criteria for the cases in your study?

These points have been clarified (lines 148-235).

Line 107: Regarding the evaluations of age: Either the age is normally distributed - in which case the mean value is given with standard deviation, or the age is not normally distributed, in which case the mean value is shown with median - minimum-maximum. The statistics are confusing. The evaluation of breeds is extremely minimalistic.

The paragraph has been clarified (lines 240-249).

Results:

Figure 1+2: Ultimately, the figures only show where the samples come from geographically in Italy. No evaluation or conclusions were drawn from the distribution. The figures are, therefore, superfluous.

The pictures provided are included to show that the cases are not exclusively sent by practitioners from Umbria even if the majority of the samples are submitted from the locals, diversifying the population. Additionally, the other reviewer did not express any comments, therefore, we would like to keep them in the manuscript.

Table 1: The proportion of 228 melanoma in the skin is surprisingly high. Usually, melanocytic tumors of the skin are benign (melanocytoma). At no point has it been explained why the proportion of melanomas in the skin is so high in this study? What classification/grading is used in the pathological institution? à Mat&Met?

The diagnostic method has been added in the material and method section (lines233-235). An explanation about the high number of melanomas has been added in the discussion section (lines 443-454).

On the other hand, the proportion of melanocytomas in the mucocutaneous area is very high. Furthermore, it is not described whether these are mucocutaneous transitions in the mouth or, for example, in the genital tract. There is also no more precise information on the cutaneous localization (e.g., back, limb).

Regarding mucocutaneous melanocytomas, we believe that 43 cases in 19 years is not exceptionally high. Furthermore, as stated in Esplin DG. (Survival of dogs following surgical excision of histologically well-differentiated melanocytic neoplasms of the mucous membranes of the lips and oral cavity. Vet Pathol. (2008) 45:889–96. doi: 10.1354/vp.45-6-889), it is recognized that a population of tumors arising from the mucous membranes of the lip and oral cavity are well differentiated and have a low progression. Moreover, depending on the study, mucocutaneous tumors are sometimes included among the oral tumors or analyzed separately. In our study, mucocutaneous tumors include lip and eyelid neoplasms and were considered in this category only tumors that were specifically “at the” mucocutaneous junction.

Regrettably, the specific information regarding the cutaneous localization was not documented in all of our records, therefore we have decided to omit this detail.

Lines 140-148: The evaluation of the genders is unsystematic and confused. Here, all melanocytic tumors are evaluated according to gender - but without any indication of significance. The distribution of gender in melanomas or melanocytomas is given without numerical information.

The numerical data and relative percentages of malignant and benign melanocytic tumors for each gender are now reported for dogs (lines 333-335) and cats (lines 406-408).

Line 149: 3.16 (point instead of comma)

The mistake has been corrected.

Line 150: Is the tumor of the 1-year-old dog a melanoma or a melanocytoma? Where was it localized, and what kind of dog was it? The same questions arise for the tumor of the oldest dog. And later, the same for the cats...

This information has been added (lines 338-339).

Line 151: See Material & Methods for the age data

The data has been corrected.

Figure 2: Are there statistically significant differences between the melanocytomas and melanomas? The representation in the boxplots refers to medians/quartiles; the text refers to mean and standard deviation - a wild mixture of different statistical data.

Yes, as reported in lines 351 age differences between melanoma and melanocytoma are significant in dogs. To keep uniformity in the manuscript, boxplots were used in both, dogs (normally distributed data) and cats (skewed distributed data). The mean and median overlap in dogs.

Line 156: The heading is in the wrong place

The mistake has been corrected.

Lines 160-166: The evaluation of breeds is only descriptive, and no reference is made to the overall population; therefore, no breed disposition can be read from this. It would be helpful to indicate the orthopaedic Joos in relation to the total population.

We apologize, but we are unfamiliar with the parameter or formula "orthopaedic Joos" that the reviewer is referring to. A Google search has not provided any relevant context; therefore we respectfully request that the reviewer elaborate on their meaning.

Regarding the predisposition and distribution of tumors across different breeds, we have opted for a purely descriptive evaluation. Primarily, this is because most tumors occur in mixed-breed animals. Additionally, there are no validated population registries in our area that would allow us to accurately assess the true incidence of a disease in a specific breed. Consequently, we cannot definitively determine whether the higher prevalence of a tumor in a particular breed is due to a true genetic predisposition or simply to a higher frequency of that breed in a specific geographic area (e.g., due to cultural factors, breed popularity, etc.).

Are there statistically significant differences between melanocytomas and melanomas? (races, age, gender)? This is not clearly elaborated.

Information was present in the text following the descriptive data. Information about the significance (p-value), that was lacking, and the statistical test performed in materials and methods is now implemented.

Melanocytic tumors of the cat: There are different stages and classifications specifically for ocular melanocytic tumors of the cat, which are not discussed in detail here. However, this would be desirable.

We are aware that feline ocular melanocytic tumors follow a specific classification, however, this is not part of the aim of our study. Similarly, in dogs, details on growth patterns and tissue organization were not reported.

Information on age, youngest and oldest cat as for dogs

This information has been added (lines 418-419)

Do you only have DSH and Persians in Italy???? – Which breeds are in your total population of pathology samples?

In the dataset of feline melanocytic tumors, these breeds are the ones represented. However, as mentioned in the text, for many cats included in the study, the breed was not specified, so we cannot know with certainty if these cases would have come from other breeds. In the overall population of samples sent to our laboratory, there are other breeds as well, but the majority remain Domestic Shorthair (DSH) cats.

Discussion:

Line: The authors write that this study would provide valuable insights into the epidemiology of the tumors. However, which information from the present survey has what value and why? - Especially as the authors note that almost all known publications come to the same results.

An explanation has been added to the paragraph (lines 455-461).

Line 225: It was previously stated that there is no significant age difference between dogs with melanoma and melanocytoma. With a standard deviation of 3 years and an age difference of 9 to 11 years, one cannot speak of a trend. Furthermore, the effects of the breed were not taken into account. To what extent do melanomas or melanocytomas occur more frequently or less frequently in certain breeds - some breeds have a longer life expectancy than others. The data is interpreted very uncritically here.

As reported in lines 351 the age was significantly different in dogs with melanoma compared to melanocytoma. We agree with the reviewer that the breed effect was not evaluated, but this was due in part to the reasons explained above and in part to the fact that it was not the main purpose of this study. However, we added a comment at lines 470-477)

Furthermore, looking at the breeds, the hair coat color should be considered!

Unfortunately, this data is not available.

Line 232 ff: Only results are repeated here, and there is no critical examination of the literature or the author's data. There is no indication of limitations and no explanation of what the study really brings that is new.

The paragraph has been implemented

Line 243: The regular oral cavity examination is not a conclusion from the present work! - This is generally known because previous studies and experiences are already available.

The statement in question was not intended to serve as a conclusion to our work. It has now been rephrased and clarified for better understanding.

Line 245 ff: So far, no recognizable regional differences between the results of the own study from Italy compared to other studies have been identified, so it is questionable why further studies should be conducted with others in catchment areas and regions. The possible predisposing factors, such as UV light, genetic factors, racial predisposition, racial distribution in the different countries, etc., are not discussed.

The idea is that involving more institutions and regions could increase the number of variables for the statistical analysis that we do not have. By doing so we could have an updated analysis of the epidemiologic landscape in Italy.

Conclusions

Line 258: The conclusion that the results of this study have any impact on prognosis or veterinary practice cannot be drawn from the results presented in this study.

The sentence has been rephrased.

Reviewer 2 Report

Comments and Suggestions for Authors

The current manuscript described the epidemiology on melanocytic tumors in canine and feline population from the veterinary pathology service center of university of Perugia. This paper is meaningful to help catch a whole picture on the relative diseases of pets and may provide some information to the same disease in human beings.

1, The author performed analysis for 845 canine MTs. Since dogs are one animal type to mimic drug effects within humans. It is good to provide some comparative analysis between dog MTs and human MTs.

2, Since the authors described the feature of MTs of canine and feline simultaneously. It is interesting to know that dogs are more likely to have cutaneous neoplasms, while feline animals are susceptible to ocular melanoma. The authors could consider adding some information why there are differences between the 2 species.

3, Regarding ocular melanoma, is it the same to uveal melanoma (UM). The UM is one of the most aggressive cancer types and is very likely to disseminate to liver tissues for human beings. There is no effective treatment available for the cancer type. The authors could tell more information to compare the disease between cats and humans.

4, The authors should provide one table to summarize the findings from different breeds of dogs and felines. The current description is provided in text and may not be concise to get the details. 

Comments on the Quality of English Language

I feel that the quality of English language is good enough for the publication.

Author Response

The current manuscript described the epidemiology on melanocytic tumors in canine and feline population from the veterinary pathology service center of university of Perugia. This paper is meaningful to help catch a whole picture on the relative diseases of pets and may provide some information to the same disease in human beings.

We are truly grateful, for the evaluation given by the reviewer. Their valuable feedback and suggestions have improved the quality and clarity of our document. Following are the response to reviewer’s comments:

1, The author performed analysis for 845 canine MTs. Since dogs are one animal type to mimic drug effects within humans. It is good to provide some comparative analysis between dog MTs and human MTs.

The information has been added on the introduction (Lines 68-79).

2, Since the authors described the feature of MTs of canine and feline simultaneously. It is interesting to know that dogs are more likely to have cutaneous neoplasms, while feline animals are susceptible to ocular melanoma. The authors could consider adding some information why there are differences between the 2 species.

To the authors’ knowledge, no specific studies have addressed the site and behavior differences among canine and feline melanocytic tumors, which is why we refrained from providing an explanation. However, if the reviewer is aware of specific scientific references we would be happy to cite them.

3, Regarding ocular melanoma, is it the same to uveal melanoma (UM). The UM is one of the most aggressive cancer types and is very likely to disseminate to liver tissues for human beings. There is no effective treatment available for the cancer type. The authors could tell more information to compare the disease between cats and humans.

The information has been added on the introduction (lines 80-88).

4, The authors should provide one table to summarize the findings from different breeds of dogs and felines. The current description is provided in text and may not be concise to get the details.

The tables has been added.

Round 2

Reviewer 1 Report

Comments and Suggestions for Authors

Dear Authors,

the manuscript has been improved markedly .

However, you can find some further comments in the pdf as comments.

Author Response

in Italy

The title has been implemented

in humans!

The line has been implemented

in contrast (instead of: compared)

The line has been corrected

Data from germany should be included in detail see your reference 33

Data has been added

Ref 33 should be included

Ref 33 has been included

You should give the numbers of missing data  - or in a positive way: make a summarizing table with signalement data correlated to the three diagnoses in cats and dogs

The missing data for sex, age and breed (lines 198, 215, 222, 256, 264, 273 ) are already reported in the results. Two additional tables with all the cases and the signalement were added in the supplementary material.

but a "certain diagnosis" was an inclusion criterium?

Yes, as reported in line 143, samples with a certain diagnosis of melanocytic tumor were included. Samples where melanocytic tumor was only included in the differential diagnosis list and not confirmed on histology or histology+IHC were excluded.

Again: I understand that you are proud on your country. However, in times of globalisation, it is not certain that the animals lived always in this region or have been born there, their parents may come from other countries and so on - thus the local effects should not be over interpreted - this is always a limitation!

The data reported in the map are not meant to report the local and environmental role on the epidemiology, but just the samples origin.

see commenty above - summarizing table would be good!

The supplementary tables has been added, as requested by the reviewer.

which is not significant!? - in the text you say it is significant but in the figures there are no *** to show this you should include the number of cases of melanocytomas and melanomas and make the legend more prrecise - the figure including legend should be self-explaning without reading the main text! position of legend of figures below the figure

The description has been implemented and the figure has been changed.

Include a row with total number of melanocytomas and melanomas and I cannot believe that then there is no significance for example in rottweilers 8 melanocytomas versus 26 melanomas. maybe - overall there is no significance but looking to the certain breeds - there probably id significance which is also well described in literature. However you do not cover all breeds, but especially rhodesian ridgebacks and weimaraner very often have melanocytomas of the skin but rarely melnaoms! - quiet in opposite to rottweilers....

We agree with the reviewer that it may seem counterintuitive that there were no statistically significant differences. However, we conducted statistical analyses on the entire population and did not find any significant results between the different breeds and the type of diagnosis. This may be due to the small sample size within individual subgroups, limiting our statistical power. We have added a sentence to the discussion highlighting this potential limitation of the study, which could explain why our findings do not perfectly align with the literature.

Regarding breed distribution, some breeds known to be predisposed to melanocytomas, such as the Rhodesian Ridgeback, were represented by very few individuals in our study. Consequently, they were grouped into the "other breeds" category.

The table has been implemented.

see comment Figure 1

As previously mentioned, our intention is not to attribute epidemiological significance to this data; rather, we want to emphasize that the samples are not exclusively from the Umbria region.

12.3%: dot not comma - see also the following lines

The mistakes have been corrected

correct the numbers - this is figure 4 and see comment to figure 2

The number has been corrected and the description implemented.

see reference 33

Line 291 has been implemented

do you think, that compared to the other studies which mainly derive from northern europe - the UV light factor may be pathogenetically relevant in your population in italy???

Actually, many previous epidemiological studies have been conducted in Northern European regions, where climatic conditions are significantly different. However, in the case of dogs and cats, we do not believe that sun exposure has a considerable impact on tumorigenesis, as the BRAF mutation commonly associated with UV-induced tumors in humans is not present in these animals, suggesting that the underlying mechanisms of tumor development in these species may be distinct and not influenced by solar radiation.

However, there is no continuum from melanocytoma to melanoma in dogs and cats! the oncogenesis seems to  be age dependent....

A comment has been added in the section.

This is the cause why other studies to multivariant regression analysis with mixed breeds as reference see Grüntzig et al. and Aupperle-Lellbach et al. and others

We fully agree on the necessity of a multivariate regression analysis. However, to carry this out effectively, it will be essential to conduct a further study in collaboration with other Italian institutions, as unfortunately, the number of cases at our single institution is not sufficient for a robust analysis.